# Unveiling The Matthew Effect Across Channels: Assessing Layer Width Sufficiency via Weight Norm Variance

**Yiting Chen, Jiazi Bu, Junchi Yan**[‡]
Dept. of CSE & School of AI & MoE Key Lab of AI, Shanghai Jiao Tong University
{sjtucyt, bujiazi001, yanjunchi}@sjtu.edu.cn
https://github.com/Ytchen981/Channel_Matthew_Effect

## Abstract

The trade-off between cost and performance has been a longstanding and critical issue for deep neural networks. One key factor affecting the computational cost is the width of each layer. However, in practice, the width of layers in a neural network is mostly empirically determined. In this paper, we show that a pattern regarding the variance of weight norm corresponding to different channels can indicate whether the layer is sufficiently wide and may help us better allocate computational resources across the layers. Starting from a simple intuition that channels with larger weights would have larger gradients and the difference in weight norm enlarges between channels with similar weight, we empirically validate that wide and narrow layers show two different patterns with experiments across different data modalities and network architectures. Based on the two different patterns, we identify three stages during training and explain each stage with corresponding evidence. We further propose to adjust the width based on the identified pattern and show that conventional layer width settings for CNNs could be adjusted to reduce the number of parameters while boosting the performance.

## 1 Introduction

The cost-accuracy trade-off has been a longstanding and critical issue for deep neural networks. As one key factor that affects the computational cost, the width of each layer is mostly empirically determined or extensively searched over an architecture space in neural network architecture search literature [43, 28]. Despite the research on over-parameterization [24, 23] and empirical evidence [37, 15, 31, 41] showing that the wider network leads to better performance, there are few works about how we allocate the computational resources across the layers. On understanding the layer width requirement in a deep neural network, [20] theoretically proves the lower bound of layer width for the model to approximate any Lebesgue-integrable function while [4] improves the width lower bound with dynamic systems. However, a practical indicator of sufficient layer width is still lacking.

In this paper, we identify a simple pattern difference between wide and narrow layers regarding the weight norm variance between different channels during training. A simple observation is that larger-weight channels would have a larger gradient. As the weight norm increases during training [30, 11], we show the disparity of weight norm between similar channels also increases during training, which we call the Matthew effect between channels. It motivates us to investigate the weight norm variance across the channels in each layer during training. With experiments on image, graph, and text datasets (including CIFAR-10 [18], cora [35], and enwiki8 [22]) with various model structures including

---

[‡]Corresponding author. This work was in part supported by NSFC (92370201, 62222607) and Shanghai Municipal Science and Technology Major Project under Grant 2021SHZDZX0102.

38th Conference on Neural Information Processing Systems (NeurIPS 2024).

(GCN [17], GRU [6], ViT [9]), we show that wide and narrow layers show two different patterns during training. As we change the layer width, we show that narrow layers show a decrease until the saturate (DS) pattern, such that the weight norm variance first increases and then decreases. For wider layers, the weight norm variance keeps increasing until saturate (IS). We conjecture that, for sufficiently wide (or over-parameterized) layers, channels with a small weight norm would always be surpassed by similar channels with larger weights, and therefore, the variance keeps increasing until convergence. On the other hand, for narrow layers with few channels, the channels with small weights could be orthogonal to channels with large weights, and the variance decreases.

We further show that the training from random initialization to convergence can be divided into three stages for neural networks showing the identified IS or DS pattern. In the first stage, typically a few epochs after random initialization, since the gradient is almost orthogonal to the weight vector, the variance between weight norms corresponding to different neurons does not increase or even decrease. In the second stage, both the performance of the network and the variance between neurons drastically increase for a dozen epochs. It indicates a fast learning process with the weight norm of a few neurons increasing drastically. In the third stage, the wide layers show the IS pattern, such that the weight norm variance keeps increasing and staying high, while the narrow layers show the DS pattern, such that the weight norm variance starts to decrease. We provide our explanations for each stage of training with corresponding empirical evidence on VGGs [29] trained on CIFAR10.

Furthermore, we show that the conventional layer width setting for CNNs such as VGG or ResNet might not be optimal. Generally, with the conventional layer width setting, the layers in the middle show the DS pattern while the former and latter layers show the IS pattern. It indicates that the layers in the middle could use more width, which also cohere with the intrinsic dimension firstly increasing then decreasing and reaching the high point at the middle [2]. We propose adjusting the width of each layer of widely used networks to boost the performance and reduce the number of parameters. **We summarize the contributions of this paper in the following:**

- To our best knowledge, we are the first to identify patterns regarding the variance of weight norm between different channels during training that can indicate whether a layer is wide enough. We have observed the pattern with different model structures on different data modalities (including CNNs and ViTs for image classification [29, 14, 9], GCN for graph node classification [17], and GRU for language modeling [6]).
- Based on the identified pattern, we further show that there are, in general, three training stages from random initialization to convergence. We provide our explanations with empirical evidence on VGG models trained for image classification that supports our explanation.
- We verify that with the identified pattern, the conventional width setting for CNNs could be adjusted for fewer parameters and better performance on CIFAR10 and TinyImageNet.

## 2  Related Works

**Researches on the Width of Neural Networks.** By simply widening the network, early works [37, 15] empirically show that the performance can be improved. On the other hand, structured pruning methods [33, 36, 16] propose to reduce the width for efficiency. Compared to the static pruning methods, another branch of research focusing on dynamically changing the width of the network conditioned on the input by skipping neurons in linear layer [3], branches in Mixture of Experts (MoE) [10] and channels in CNN [12, 19] at inference time. Readers interested in dynamic networks could refer to [13] for more details. Notably, a recent work [5] proposes to merge neurons to reduce the computational cost where layers at the middle have fewer neurons that could be merged. Similar results are also reported regarding investigating the intrinsic dimension [2] across the layers. Unlike these previous works focusing on the trained model, we show that one could identify the

| Phenomenon | Description | Measured by |
|---|---|---|
| The critical period in training [1] | Providing corrupted data in the early training phase leads to irrevocable damage to final performance | Testing accuracy after convergence. |
| Frequency-principle [27, 34] | Neural networks first learn low-frequency information and then high-frequency information. | Gradient norm on different frequency components. |
| Grokking [26] | Neural networks show a period of near-perfect training performance and nearly random guessing test performance before generalization. | Training and testing accuracy. |
| Double-descent [23] | The testing accuracy firstly improves, then worsens, and then eventually improves as the model capacity increases or the training goes on. | Testing accuracy. |
| Ours | See details in Fig. 1(b). | Weight norm variance in a layer |

Table 1: We list some different training dynamics phenomena.

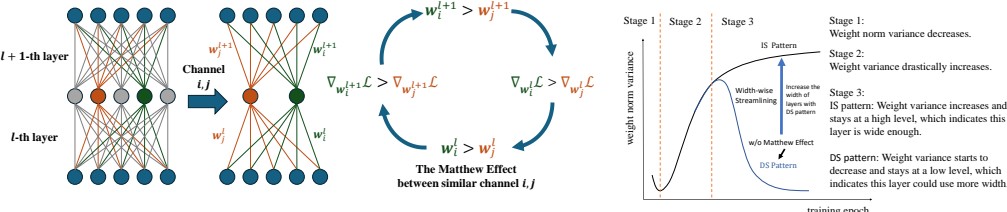

(a) Illustration of the Matthew effect between channels with weight vectors of similar direction.

(b) Two different patterns we identified of how weight norm variance change during training regarding layer width

Figure 1: Illustration of our motivation and the identified weight norm variance pattern. **Fig. (a)**: For two channels with weight vectors of similar direction, the gradient of the former layer is proportional to the weight norm of the latter layer and vice versa. As empirical evidence indicated, the weight norm increases during training, and the weight norm disparity between channels increases. **Fig. (b)**: The two different patterns we identified of how weight norm variance would change during training. For wide layers, the weight norm variance keeps increasing, which we call the increase to saturate pattern (IS). For narrow layers, the weight norm starts to decrease after reaching a high point, which we call the decrease to saturate pattern (DS).

layers with insufficient width during training. We further propose a simple width-wise streamlining technique to boost the network performance and reduce the number of parameters. Note that the objective of our width-wise streamlining technique differs from the pruning method and we make the FLOPs of the adjusted model close to the original network.

**Dynamics of Network Training** Investigating the training process of neural networks has always been an active research topic. The early phase of neural network training is investigated in [11] and a timeline for the early phase of training (the first several epochs) is provided while other works [24, 8, 25] have focused on the later training phase investigating the effect of over-parameterization on the generalization and the training heuristics that lead to simpler solutions. From the frequency spectrum perspective, F-principle [27, 34, 32] shows that low-frequency components play a more important role in the generalization and that the training of CNN could be divided into two phases where the CNN firstly learn low-frequency components then learn high-frequency components. The frequency principle theoretically proves that the gradient on low-frequency components will be larger [21]. In this paper, we analyze the training dynamics from the weight norm variance perspective and show that the training of neural networks could be summarized into three stages in general. To our knowledge, this is the first work to investigate the variance between weight norms corresponding to different neurons. We list some well-known phenomena in Table 1. While a recent work [7] correlates grokking [26] with double-descent [23], we hope the relationship between the three stages of training and the other phenomena could be further explored in future works. For example, the difference between the second stage and the third stage could be the result of network learning different frequency components, easy or hard data samples [42] *etc*.

## 3 Different Patterns between Wide and Narrow Layers

In this section, we first provide a simple analysis showing the motivation for us to investigate the weight norm variance across the channels of a layer. We then provide empirical results from experiments across different data modalities and model structures to show that there are two different patterns between wide and narrow layers.

### 3.1 Simple Analysis Regarding Channels in a Layer During Training

Let us consider an MLP with ReLU activation for simplicity. Suppose we have $d$ channels in the $l$-th layer. Let $\mathbf{z}^{l-1} \in \mathbb{R}^{d_{in}}$ and $\mathbf{z}^{l+1} \in \mathbb{R}^{d_{out}}$ denote the features at the $l-1$-th layer and the $l+1$-th layer. The weight for $l$-th layer and $l+1$-th layer is $W^l \in \mathbb{R}^{d_{in} \times d}$ and $W^{l+1} \in \mathbb{R}^{d \times d_{out}}$. Then we have:

$$\mathbf{z}^{l+1} = W^{l+1}\sigma(W^l\mathbf{z}^{l-1}) \tag{1}$$

where $\sigma(\cdot)$ is the ReLU activation function. Let $\mathbf{w}_i^l \in \mathbb{R}^{d_{in}}$ denotes the $i$-th row vector of $W^l$ and $\mathbf{w}_i^{l+1} \in \mathbb{R}^{d_{out}}$ denotes the $i$-th column vector of $W^{l+1}$, the output corresponding to the $i$-th channel is $\sigma(\mathbf{w}_i^{l\top}\mathbf{z}^{l-1}) \cdot \mathbf{w}_i^{l+1}$ and the output $z^{l+1}$ is the combination of the outputs of all the $d$ channels as

$$\mathbf{z}^{l+1} = \sum_{i=1}^{d} \sigma(\mathbf{w}_i^{l\top}\mathbf{z}^{l-1}) \cdot \mathbf{w}_i^{l+1} \tag{2}$$

Suppose the loss function is $\mathcal{L}$, and the gradient for $\mathbf{z}^{l+1}$ is $\nabla_{\mathbf{z}^{l+1}}\mathcal{L}$, then we have

$$\nabla_{\mathbf{w}_i^{l+1}}\mathcal{L} = \begin{cases} \mathbf{w}_i^{l\top}\mathbf{z}^{l-1} \cdot \nabla_{\mathbf{z}^{l+1}}\mathcal{L}, & \mathbf{w}_i^{l\top}\mathbf{z}^{l-1} \geq 0 \\ \mathbf{0}, & \mathbf{w}_i^{l\top}\mathbf{z}^{l-1} < 0 \end{cases} \tag{3}$$

$$\nabla_{\mathbf{w}_i^l}\mathcal{L} = \begin{cases} (\mathbf{w}_i^{l+1\top}\nabla_{\mathbf{z}^{l+1}}\mathcal{L}) \cdot \mathbf{z}^{l-1}, & \mathbf{w}_i^{l\top}\mathbf{z}^{l-1} \geq 0 \\ \mathbf{0}, & \mathbf{w}_i^{l\top}\mathbf{z}^{l-1} < 0 \end{cases} \tag{4}$$

For two channels $m, n \in [1, d]$, suppose $\frac{\mathbf{w}_m^{l+1}}{\|\mathbf{w}_m^{l+1}\|} \approx \frac{\mathbf{w}_n^{l+1}}{\|\mathbf{w}_n^{l+1}\|}$ and $\frac{\mathbf{w}_m^l}{\|\mathbf{w}_m^l\|} \approx \frac{\mathbf{w}_n^l}{\|\mathbf{w}_n^l\|}$ then $\mathbf{w}_m^{l\top}\mathbf{z}^{l-1}$ and $\mathbf{w}_n^{l\top}\mathbf{z}^{l-1}$ would have the same sign and we have

$$\nabla_{\mathbf{w}_m^{l+1}}\mathcal{L} \approx \frac{\|\mathbf{w}_m^l\|}{\|\mathbf{w}_n^l\|}\nabla_{\mathbf{w}_n^{l+1}}\mathcal{L}. \tag{5}$$

$$\nabla_{\mathbf{w}_m^l}\mathcal{L} \approx \frac{\|\mathbf{w}_m^{l+1}\|}{\|\mathbf{w}_n^{l+1}\|}\nabla_{\mathbf{w}_n^l}\mathcal{L}. \tag{6}$$

It means that for channels with weights of similar direction, the larger $\|\mathbf{w}^l\|$ leads to larger $\nabla_{\mathbf{w}^{l+1}}\mathcal{L}$ and the larger $\|\mathbf{w}^{l+1}\|$ leads to larger $\nabla_{\mathbf{w}^l}\mathcal{L}$. While empirical evidence in previous works [30, 11] has shown that the weight norm increases during training. It means that, for two channels with similar weight direction, the weight norm disparity between them would generally keep increasing during training, which we call **the Matthew effect between similar channels**. For the channel with a larger weight, the larger weight norm at the $l+1$ layer leads to a larger gradient of the corresponding weight at the $l$ layer, which increases the weight norm at the $l$ layer. In turn, a larger weight norm at the $l$ layer further speeds up the increase of the weight norm at the $l+1$ layer. According to [5], sufficiently wide layers may learn many similar channels after training, therefore, we expect the variance of weight norm across different channels would be higher in the wide layer than in the narrow layer. It motivates us to investigate further the weight norm variance between different channels in a layer.

### 3.2 Empirical Evidence of Two Different Patterns for Wide and Narrow Layers.

In this subsection, we investigate the variance of weight norms corresponding to different channels. For the $i$-th channel in Eq. 2, with ReLU as activation functions we have

$$\sigma(\mathbf{w}_i^{l\top}\mathbf{z}^{l-1}) \cdot \mathbf{w}_i^{l+1} = \begin{cases} \mathbf{w}_i^{l+1}(\mathbf{w}_i^{l\top}\mathbf{z}^{l-1}), & \mathbf{w}_i^{l\top}\mathbf{z}^{l-1} \geq 0 \\ \mathbf{0}, & \mathbf{w}_i^{l\top}\mathbf{z}^{l-1} < 0 \end{cases} \tag{7}$$

Therefore, we define the weight norm corresponding to the $i$-th channel as $\|\mathbf{w}_i^l\| \cdot \|\mathbf{w}_i^{l+1}\|$[1]. We conduct our experiments on several datasets across different data modalities with different model architectures and record the variance of corresponding weight norms during training.

We report the weight norm variance results for Graph Convolutional Neural Network (GCN) [17] trained on graph dataset cora [35], Gated Recurrent Units (GRU) [6] trained on text dataset en-wiki8 [22] and Vision Transformers (ViT) [9] trained on image dataset CIFAR10 [18]. As we change the width of a certain layer (*e.g.* the MLP module in the transformer), we show that the wide layer and the narrow layer show two different patterns. For the wide layer, the weight norm variance firstly increases and stays at a high level until convergence, which we call the increase to saturate (IS) pattern. For the narrow layer, the weight norm variance decreases after the initial increase, which we call the decrease to saturate (DS) pattern.

---

[1]It is similar for other ReLU-like activation functions such as leaky ReLU.

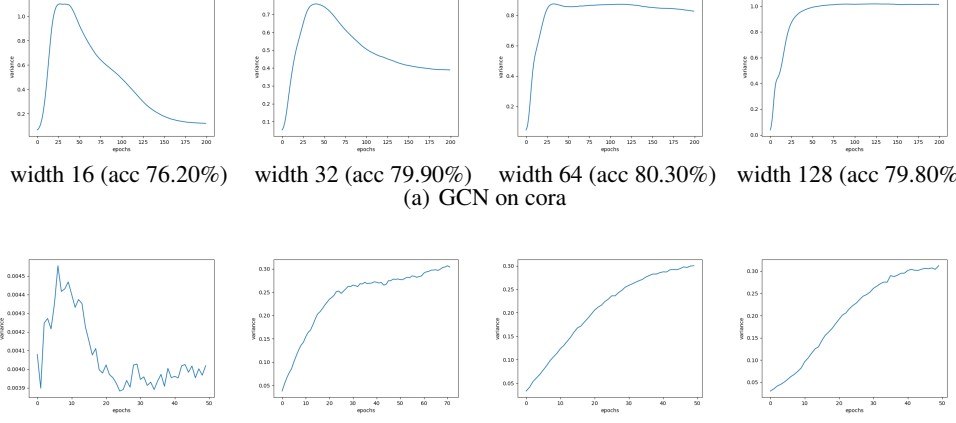

width 16 (acc 76.20%)   width 32 (acc 79.90%)   width 64 (acc 80.30%)   width 128 (acc 79.80%)

(a) GCN on cora

input-hidden, width 500  input-hidden, width 800  input-hidden width 1000 input-hidden width 1150

(b) GRU on enwiki8

Figure 2: Weight norm variance change during training of GCN [17] on cora [35] and GRU [6] on enwiki8 [22]. We change the width of the second layer in GCN and the input-hidden weight of the second layer in GRU. As the width increases, the weight norm variance during training shows two different patterns. For the wide layer, the weight norm variance increases and stays at a high level until convergence, which we call an increase to saturate (IS) pattern. For the narrow layer, the weight norm variance decreases after the initial increase, which we call the decrease to saturate (DS) pattern.

As shown in Fig. 2, we present the results of GCN [17] on cora [35] and GRU [6] on enwiki8 [22]. In Fig. 2(a), the results show the weight norm variance change of the second layer with different widths. As the width increases from 16 to 64, the weight norm variance changes from DS pattern to IS pattern. Notably, increasing the width when the DS pattern is identified (from 16 to 64) leads to an accuracy boost, while increasing the width when the IS pattern is identified (from 64 to 128) does not. In Fig. 2(b), we report the results of the weight norm variance change of the input-hidden gate of the second layer of the GRU. Though the recurrent neural networks (RNN) have different architectures compared to the multi-layer ReLU network we discussed before, the weight norm variance of the narrow and wide layers also shows DS and IS patterns, respectively.

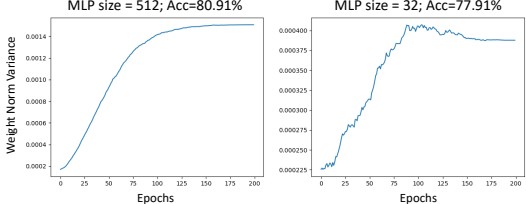

Figure 3: Weight norm variance between different channels of the MLP at the 5th layer of a tiny ViT trained on CIFAR-10. The left figure corresponds to the result where the MLP size is set as 512, and the right figure corresponds to the result where the MLP size is set as 32.

Fig. 3 shows the results of two ViTs trained on CIFAR10. We use a ViT with 6 layers where each layer has 8 heads and the hidden size for each attention layer is 512. The width of the MLP at the 5th layer is set at 32 and 512 respectively. For the layer of width 512, the weight norm variance always increases and stays at a high level following the IS pattern. As we reduce the width of the MLP to 32, the weight norm variance starts to decrease after reaching a high point following the DS pattern. For more details on the GCN, GRU, and ViT training, please refer to Appendix B.

**Our Conjecture About the Different Patterns Regarding Layer Width.** As shown in Eq. 2, each layer is like a "mixture of experts" where each channel has the same weight. The only difference between channels is the random initialization. For wide layers with a large number of channels, the chance for two channels learning weight of similar direction is much higher than that for narrow layers with limited channels. Therefore, the weight norm variance among channels of a wide layer does not decrease while the weight norm variance among channels of a narrow layer decreases after reaching a high point. In the next section, we provide a more detailed analysis with empirical evidence about the three training stages we identified.

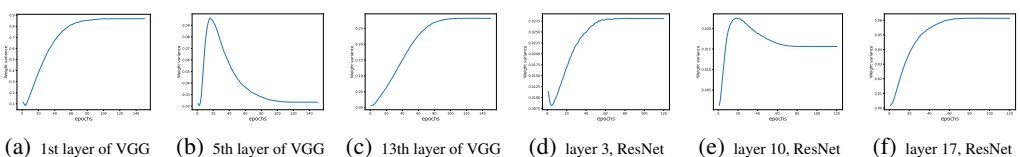

| (a) 1st layer of VGG | (b) 5th layer of VGG | (c) 13th layer of VGG | (d) layer 3, ResNet | (e) layer 10, ResNet | (f) layer 17, ResNet |

Figure 4: The variance of weight norm $\|\mathbf{w}_i^l\| \cdot \|\mathbf{w}_i^{l+1}\|$ at different layers of VGG-16 and ResNet18 trained on CIFAR-10. The y-axis corresponds to the variance while the x-axis corresponds to the epochs trained after initialization. From left to right is the result for the first layer, the layer in the middle, and the last layer. For more results please refer to Appendix A

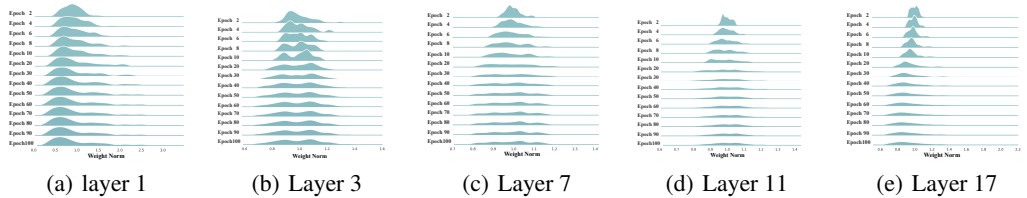

| (a) layer 1 | (b) Layer 3 | (c) Layer 7 | (d) Layer 11 | (e) Layer 17 |

Figure 5: Density plots of the weight norm of different layers of a ResNet-18 trained on CIFAR10. The x-axis corresponds to the weight norm, and the y-axis corresponds to the density. For clarity, we omit the label for the y-axis. From top to bottom, each density plot corresponds to different epochs, we demonstrate how the weight norm distribution of different layers changes during training.

## 4 Three training Stages from Random Initialization to Convergence

Based on the two patterns we identified regarding the weight norm variance, we further provide our explanation for this phenomenon. We show that, generally, we can divide the training of the neural networks showing the identified pattern into three stages. We provide our explanation for each stage with corresponding empirical evidence on VGG-16 [29] and ResNet-18 [14] trained on CIFAR10 [18]. As shown in Fig. 4, we report the weight norm variance of some of the layers in VGG-16 and ResNet-18. In Fig. 5, we also present the density plots of the weight norm for different layers of a ResNet-18 trained on CIFAR10 at different epochs. Generally, different layers show different patterns. The layers in the middle typically follow the DS pattern and the former and latter layers follow the IS pattern. We will discuss this phenomenon in the next section Sec. 5.

### 4.1 Stage 1: Weight Variance Stay Low Shortly after Initialization

Starting from a random initialization, the first stage of neural network training lasts for a few epochs. In this stage, the variance of the weight norm between neurons does not increase. At certain layers, the variance even decreases between neurons. Starting from random initialization, due to the high-dimensional weight vectors of each neuron, the gradient is more likely to be orthogonal to the weight vectors corresponding to different neurons. When measuring the cosine similarity between the gradient and the weight vector across all the channels across all the layers in a VGG16_bn trained on CIFAR-10, the largest absolute value is $0.0003$ at initialization, which provides an explanation for the weight variance not increasing.

In some scenarios, we also find weight norm variance decreases at the first stage. We conjec-

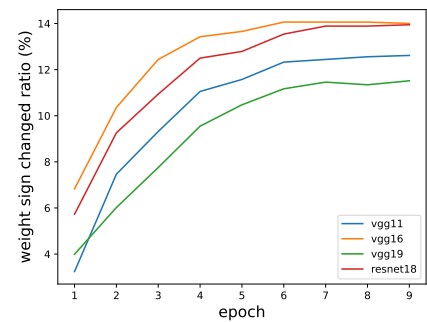

Figure 6: Ratio of weight elements where the sign is changed compared to the initialization during training different models on CIFAR-10. We show the result for the first 10 epochs. After the first 5 epochs, the sign of at least $10\%$ weights changes, indicating a dramatic change in the weight space.

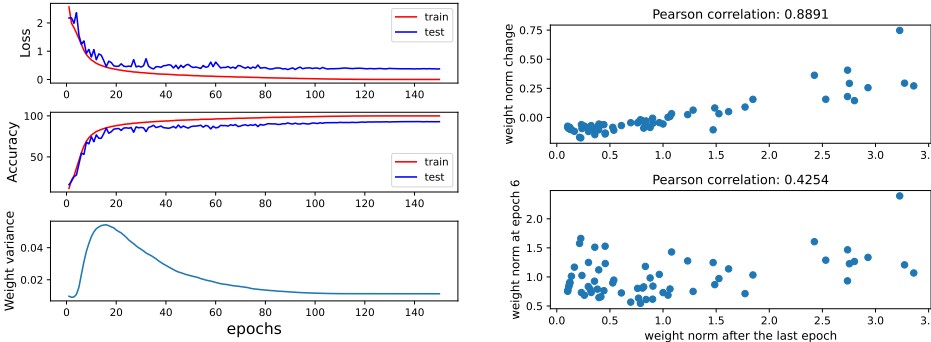

(a) The loss, accuracy, and weight norm variance of the 5-th layer of a VGG16 trained on CIFAR-10

(b) Correlation between the weight norm at the second stage and the weight norm after convergence

Figure 7: **(a)**: The loss, accuracy, and weight norm variance between neurons at 5-th layer of a VGG16 trained on CIFAR-10. The model is trained for $150$ epochs, after the second stage of training mentioned in Sec. 4.2 the model has achieved a high performance at around $20$ epoch. For more details please refer to Appendix B. **(b)**: We demonstrate the correlation between the weight norm at the 6-th epoch and the weight norm after the last epoch. The x-axis corresponds to the weight norm after the last epoch (150-th epoch). For the upper plot, the y-axis corresponds to the change of weight norm from 6-th epoch to 4-th epoch. For the lower plot, the y-axis corresponds to the weight norm at the 6-th plot. Each point represents a neuron at the first layer of VGG16.

ture that it results from the fact that the direction of weight vectors is drastically changed in the first stage. In fact, the first stage of neural network training is an extremely chaotic stage where the direction of each weight vector is dramatically changed. Fig. 6 shows that, for different networks, the sign of at least $10\%$ weight elements changes after the first 5 epochs. As a result of the dramatic change in weight space, the direction of many weight vectors is reversed and the weight norm firstly decreases and then increases.

## 4.2 Stage 2: Both Performance and Variance Drastically Increase

In the second stage, the variance of weight norm among channels starts to increase drastically. This phenomenon indicates that for some of the neurons, whose corresponding weight vectors are close to the gradient direction, the corresponding weight norm increases much faster than other neurons. At the same time, the performance also increases rapidly as shown in Fig. 7(a). The test performance after this stage is very close to the final performance, *e.g.* the testing accuracy for the VGG16 on CIFAR-10 at 20 epoch in Fig. 7(a) is at $84.65\%$, which is relatively close to the final testing accuracy $92.94\%$. We conjecture that to reach a fair performance, the network only needs to learn a few significant features, channels whose weight direction is close to the direction of the gradient would have advantages.

The second stage generally determines the relative weight norm after convergence. Compared to the chaotic first stage, the second stage is much more stable, where the weight norm of those neurons corresponding to the "significant features" discussed above constantly increases faster than other neurons. Notably, at the beginning of the second stage, the tendency of how the weight norm would change provides more information than the weight norm itself. As shown in Fig. 7(b), we demonstrate the weight norm at the beginning of the second stage and the weight norm after convergence. The result is from channels at the first layer of a VGG16 trained on CIFAR-10. The x-axis corresponds to the weight norm after convergence. For the upper subplot, the y-axis corresponds to the difference between the weight norm at the $4$-th epoch and the weight norm at the $6$-th epoch. For the lower subplot, the y-axis corresponds to the weight norm at the 6-th epoch. The Pearson correlation between the weight norm difference and the final weight norm is $0.8891$ while the Pearson correlation between the weight norm at 6-th epoch and the final weight norm is only $0.4254$. We conjecture that the channels whose weight norm increases rapidly in this stage correspond to those easy features such as the low-frequency information in Frequency-principle [27, 34, 32].

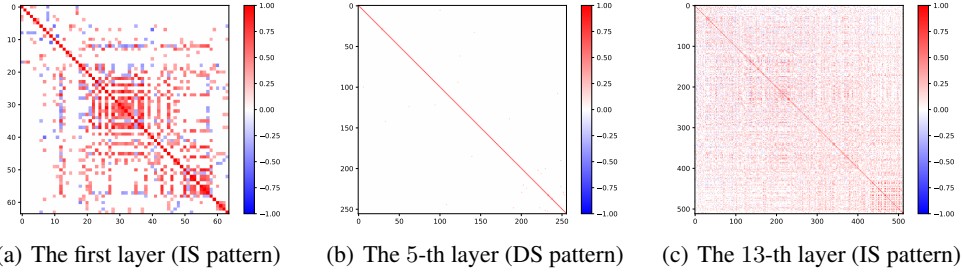

| (a) The first layer (IS pattern) | (b) The 5-th layer (DS pattern) | (c) The 13-th layer (IS pattern) |

Figure 8: The cosine similarity between weight vectors at the same layer of a VGG16 trained on CIFAR-10. The value at $i$-th row and $j$-th column correspond to the cosine similarity between the weight vectors of the $i$-th neuron and the $j$-th neuron. Note that we sort the neurons by their weight norm in descending order which means the 0-th neuron is the neuron with the largest weight norm. The results show that weights for neurons at the 5-th layer of VGG16 are almost orthogonal.

### 4.3 Stage 3: Variance Decreases or Increases to Saturation at Different Layers

The third stage is the longest stage where the performance slowly increases. In the third stage, layers show two distinct patterns which indicates whether this layer learns similar neurons. For the variance of weight norm between neurons, there are two different patterns across the layers:

- **Increase to Saturate (IS):** The variance continues to increase slowly and stay high.

- **Decrease to Saturate (DS):** The variance decreases after reaching a high value.

Interestingly, as shown in Table 2, we find that for layers where neurons (or channels) could be merged using the algorithm proposed in [5], the variance between neurons follows the IS pattern that increases and stays at a high level. On the other hand, for layers where no neurons could be merged, the variance between neurons follows the DS pattern and decreases. Fig. 8 shows the cosine similarity of weight vectors between different neurons. For layers following the DS pattern, the cosine similarity of weight vectors between most different neurons is almost zero while the cosine similarity between neurons for layers following the IS pattern is much higher. It indicates that the weight vectors of channels in layers showing the DS pattern are nearly orthogonal while layers following the IS pattern contain many similar channels.

| Layer | Width | | Third stage pattern |
| | origin | after neuron merging | |
|---|---|---|---|
| Layer 1 | 64 | 42 | IS pattern |
| Layer 2 | 64 | 63 | DS pattern |
| Layer 3 | 128 | 128 | DS pattern |
| Layer 4 | 128 | 128 | DS pattern |
| Layer 5 | 256 | 256 | DS pattern |
| Layer 6 | 256 | 256 | DS pattern |
| Layer 7 | 256 | 256 | DS pattern |
| Layer 8 | 512 | 504 | IS pattern |
| Layer 9 | 512 | 500 | IS pattern |
| Layer 10 | 512 | 509 | IS pattern |
| Layer 11 | 512 | 506 | IS pattern |
| Layer 12 | 512 | 420 | IS pattern |
| Layer 13 | 512 | 250 | IS pattern |

Table 2: We employ IFM [5] to merge the neurons (channels) across the layers of a VGG16 trained on CIFAR-10. We list the original width, the width after merging, and the weight norm variance pattern in the third stage.

Note that the neurons are sorted by the weight norm where the 0-th neuron is the neuron with the largest weight norm. Fig. 8(a) and Fig. 8(c) show that neurons with different weight norms could be in a similar direction. Further results of the weight norm distribution are presented in Appendix A.

## 5 Width Modification on CNNs for Fewer Parameters and Better Performance

As discussed in Sec. 4.3, the layers in VGG-16 show different patterns. In this section, we investigate the weight norm variance change during the training of CNNs. We empirically show that for widely used CNNs such as VGG and ResNet, the middle layers show the DS pattern and other layers

Table 3: In this table, we report the number of parameters, FLOPs, and the testing accuracy of the original VGG and ResNet models and our width-adjusted models. Each result is averaged over 10 runs with a different random seed. For a fair comparison, we make the FLOPs of the width-adjusted models close to the original model with nearly $40\%$ parameters reduced.

| Model | | Params | % of params | FLOPs | % of FLOPs | Top-1 Accuracy | |
|---|---|---|---|---|---|---|---|
| | | | | | | CIFAR 10 | CIFAR-100 |
| VGG16 | origin | 14.73M | - | 314.31M | - | $91.02 \pm 0.35$ | $66.17 \pm 0.71$ |
| | streamline width | 9.06M | 61.55% | 325.55M | 103.89% | $93.41 \pm 0.15$ | $68.63 \pm 0.31$ |
| VGG19 | origin | 20.04M | - | 399.35M | - | $91.84 \pm 0.46$ | $65.53 \pm 0.66$ |
| | streamline width | 11.50M | 57.39% | 404.95M | 101.40% | $92.38 \pm 0.09$ | $66.99 \pm 0.52$ |
| ResNet18 | origin | 11.17M | - | 557.89M | - | $92.99 \pm 0.23$ | $72.84 \pm 0.27$ |
| | streamline width | 6.26M | 55.99% | 604.22M | 108.30% | $93.60 \pm 0.13$ | $73.38 \pm 0.30$ |
| ResNet50 | origin | 23.52M | - | 1311.59M | - | $92.47 \pm 0.43$ | $73.23 \pm 0.45$ |
| | streamline width | 14.91M | 63.40% | 1404.87M | 107.11% | $93.12 \pm 0.33$ | $73.53 \pm 0.50$ |

show the IS pattern. We then adjust the layer width for VGG and ResNet, reducing the number of parameters while boosting the performance compared to the conventional layer width setting.

## 5.1 Investigating Weight Norm Variance Across the Layers of CNNs

As shown in Fig. 4, we report the variance of weight norm $\|\mathbf{w}_i^l\| \cdot \|\mathbf{w}_i^{l+1}\|$ at different layers of VGG-16 and ResNet18 trained on CIFAR-10. Generally, we find that the first several layers follow the IS pattern while the layers in the middle follow the DS pattern, and the layers at last follow the IS pattern again. It reflects the intrinsic nature of the neural network training such that layers in the middle learn orthogonal neurons. One may naturally associate this phenomenon with the results that intrinsic dimension firstly increases then decreases and reaches a high point in the middle [2]. Intrinsic dimension describes the minimum dimension needed to solve a certain problem to a certain precision level. We conjecture that intrinsic dimension is a possible explanation for the two different kinds of layers where high intrinsic dimension leads to orthogonal weights. For more results, please refer to Appendix A. It further indicates that the conventional layer width setting for CNNs might not be optimal. In the conventional setting (such as in VGGs and ResNets) the layer width increases through the first several layers and then stays large through the following layers. Since the layers in the middle could use more width and the layers in the front and back could use less width, a better setting might be that the width first increase and then decrease across the layers with the widest layer in the middle just like a streamline.

## 5.2 Streamlining the Width of Popular CNNs

To verify that setting the middle layer to be the widest is better than setting the last layer to be the widest, we adjust the width of each layer for widely used VGG and ResNet networks. For both VGG and ResNet, the width starts at 64 and ends at 512. We increase the width of the layers showing the DS pattern and decrease the width of the last several layers showing the IS pattern. For a fair comparison, we control the number of FLOPs of the width-adjusted model to be similar to the original model. As a result of the larger feature map in the middle compared

| Model on TinyImageNet | | Validation accuracy |
|---|---|---|
| VGG16 | origin | $49.78 \pm 0.92$ |
| | streamline width | $54.40 \pm 0.38$ |
| ResNet18 | origin | $54.46 \pm 0.41$ |
| | streamline width | $56.56 \pm 0.27$ |

Table 4: The result of VGG16 and ResNet18 trained on Tiny-ImageNet. Each result is averaged over 10 runs. Each model is trained for 90 epochs. More details are in Appendix B.

to the last several layers, the number of parameters is reduced by nearly half. We train these models on CIFAR-10, CIFAR-100, and TinyImageNet. Each model was trained for 120 epochs with SGD. The learning rate is 0.1 at the beginning and is reduced by 0.1 every 40 epochs. The weight decay is set at $1e-4$, and momentum is 0.9. For more details about the training recipe and the width-adjusted architecture please refer to Appendix B.

As shown in Table 3 and Table 4, we report the experimental results of our adjusted version of VGG and ResNet on CIFAR-10, CIFAR-100, and Tiny-ImageNet. Each result is averaged over 10 runs.

The results show that with similar FLOPs, by adjusting the width across the layer, we could boost the performance and largely reduce the parameters.

## 6 Conclusion

In this paper, we propose to investigate the weight norm variance among channels in the same layer and identify two different patterns between narrow and wide layers with much empirical evidence (Sec. 3). We further show that training neural networks with the identified pattern could be divided into three stages from random initialization to convergence. We provide our explanations of the three stages with corresponding empirical evidence (Sec. 4). Using the identified pattern as an indicator, in Sec. 5, we propose to adjust the conventional layer width setting of VGG and ResNet to reduce the number of parameters and boost the performance. The main limitation of the width adjustment is that we manually adjust the layer width to control the FLOPs of the adjusted model. Another limitation is that we can not determine which pattern (IS or DS) the layer follows until convergence. A more automatic and efficient width adjustment method could be proposed regarding the new weight norm variance perspective, which we leave for future works. In assessing the potential broader impact, this work provides a new perspective to evaluate the layer width setting of deep neural networks, which have the potential to advance the neural network architecture design. This work has no significant negative potential impact.

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

# A  More Results

## A.1  The Weight Norm Variance of Each Layer During Training

In this section, we present the weight norm variance $Var(\|\mathbf{w}_i^l\| \cdot \|\mathbf{w}_i^{l+1}\|)$ of each layer of VGG16, ResNet18 during training.

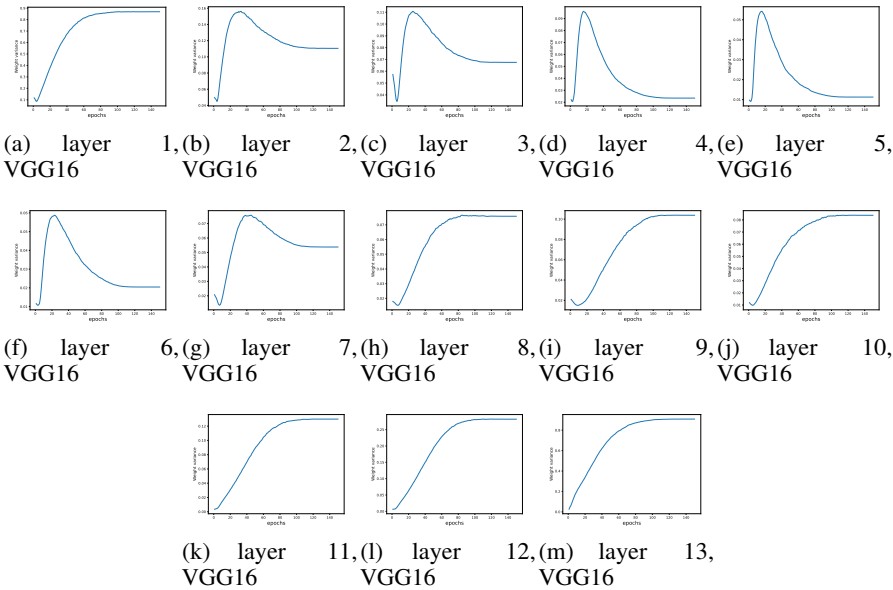

(a) layer 1, VGG16  (b) layer 2, VGG16  (c) layer 3, VGG16  (d) layer 4, VGG16  (e) layer 5, VGG16

(f) layer 6, VGG16  (g) layer 7, VGG16  (h) layer 8, VGG16  (i) layer 9, VGG16  (j) layer 10, VGG16

(k) layer 11, VGG16  (l) layer 12, VGG16  (m) layer 13, VGG16

Figure 9: The weight norm variance between neurons at each layer of VGG16 during its training on CIFAR10.

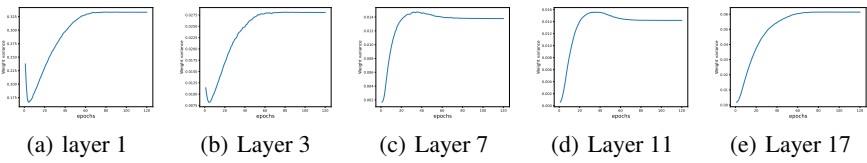

(a) layer 1  (b) Layer 3  (c) Layer 7  (d) Layer 11  (e) Layer 17

Figure 10: The results of weight variance for ResNet18.

## A.2  The weight Norm Distribution of Each Layer

In this section, we provide the distribution of $\|\mathbf{w}_i^l\| \cdot \|\mathbf{w}_i^{l+1}\|$ at each layer during training. As shown in Fig. 11, the layers following the IS pattern and DS pattern show different norm distributions.

## A.3  Cosine Similarity between Neurons

We provide the cosine similarity between the weight vectors corresponding to different neurons at the same layer in Fig. 12.

## A.4  Results with small ResNet-20

We report weight norm variance change during training of ResNet-20 and widened ResNet-20 following WRN. ResNet-20 has three blocks, with each block containing 6 layers. Since ResNet-20 is a relatively small network when it is trained on CIFAR10, the first 15 layers follow the DS pattern, indicating the layers require more width, and only the last 5 layers show the IS pattern.

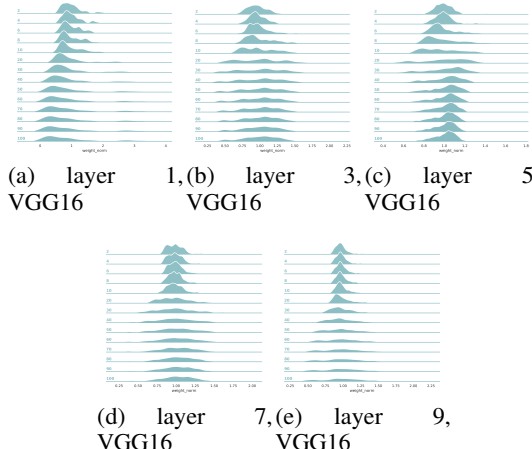

(a) layer 1, VGG16 (b) layer 3, VGG16 (c) layer 5, VGG16

(d) layer 7, VGG16 (e) layer 9, VGG16

Figure 11: The weight norm distribution of each layer of VGG16 trained on CIFAR10. The number on the left indicates the epoch.

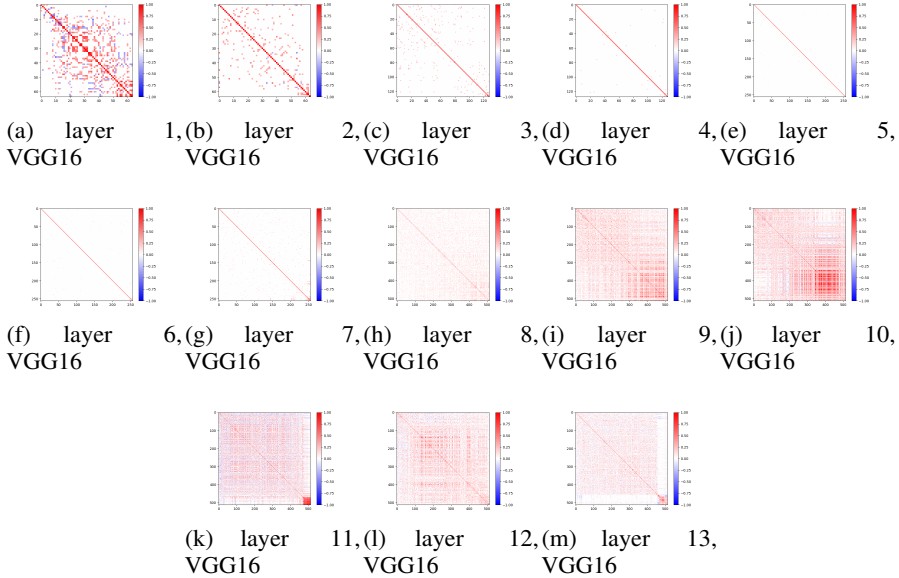

(a) layer 1, VGG16 (b) layer 2, VGG16 (c) layer 3, VGG16 (d) layer 4, VGG16 (e) layer 5, VGG16

(f) layer 6, VGG16 (g) layer 7, VGG16 (h) layer 8, VGG16 (i) layer 9, VGG16 (j) layer 10, VGG16

(k) layer 11, VGG16 (l) layer 12, VGG16 (m) layer 13, VGG16

Figure 12: The cosine similarity of weight vectors corresponding to different neurons at each layer of VGG16 trained on CIFAR10.

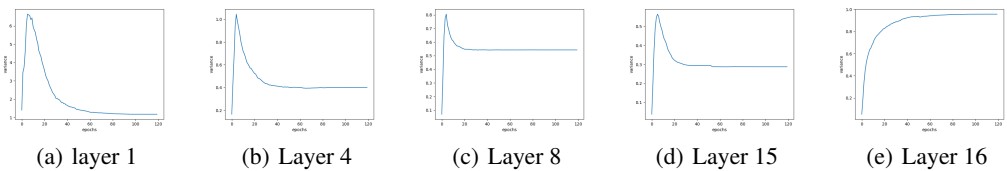

(a) layer 1 (b) Layer 4 (c) Layer 8 (d) Layer 15 (e) Layer 16

Figure 13: The results of weight variance for ResNet20.

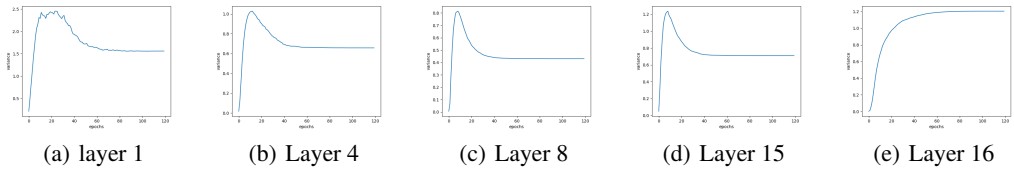

| (a) layer 1 | (b) Layer 4 | (c) Layer 8 | (d) Layer 15 | (e) Layer 16 |

Figure 14: The results of weight variance for ResNet20x4.

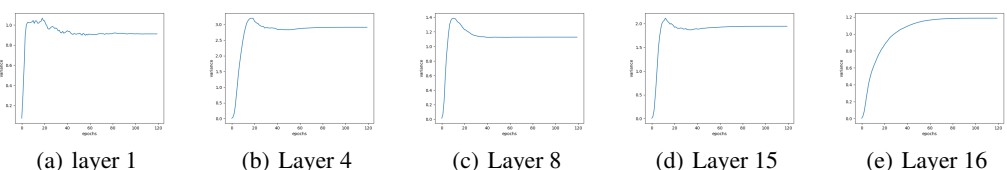

| (a) layer 1 | (b) Layer 4 | (c) Layer 8 | (d) Layer 15 | (e) Layer 16 |

Figure 15: The results of weight variance for ResNet20x8.

# B Experimental Details

## B.1 Details of Network Structure after Our Width-wise Streamlining Technique

In this section, we report the width of each model after applying our width-wise streamlining technique. We change the width of each layer of VGG and ResNet models. The width of each layer is manually set to keep the FLOPs similar to the original model. Table 5 shows the detailed width of each layer in VGG16, VGG19, ResNet18, and ResNet50.

| | | |
|---|---|---|
| VGG16 | original width | [64, 64, 128, 128, 256, 256, 256, 512, 512, 512, 512, 512, 512] |
| | adjusted width | [64, 96, 128, 160, 256, 256, 320, 512, 384, 384, 320, 256, 256] |
| VGG19 | original width | [64, 64, 128, 128, 256, 256, 256, 256, 512, 512, 512, 512, 512, 512, 512, 512] |
| | adjusted width | [64, 96, 128, 160, 256, 256, 320, 320, 512, 384, 384, 320, 320, 256, 256, 256] |
| ResNet18 | original width | [64, 64, 64, 64, 64, 64, 128, 128, 128, 128, 128, 256, 256, 256, 256, 256, 512, 512, 512, 512] |
| | adjusted width | [64, 64, 64, 64, 96, 64, 64, 160, 128, 128, 192, 128, 128, 320, 256, 256, 384, 256, 256, 256, 256, 256, 256] |
| ResNet50 first 8 blocks | original width | [64, 64, 64, 256, 256, 64, 64, 256, 256, 64, 64, 256, 256, 64, 64, 256, 256, 128, 128, 512, 512, 128, 128, 512, 512, 128, 128, 512, 512, 128, 128, 512, 512, 256, 256, 1024] |
| | adjusted width | [64, 64, 64, 256, 256, 64, 64, 256, 256, 96, 96, 256, 256, 128, 128, 512, 512, 128, 128, 512, 512, 160, 160, 512, 512, 192, 192, 512, 512, 256, 256, 1024] |
| ResNet50 last 8 blocks | original width | [1024, 256, 256, 1024, 1024, 256, 256, 1024, 1024, 256, 256, 1024, 1024, 256, 256, 1024, 1024, 256, 256, 1024, 1024, 512, 512, 2048, 2048, 512, 512, 2048, 2048, 512, 512, 2048] |
| | adjusted width | [1024, 320, 320, 1024, 1024, 384, 384, 1024, 1024, 320, 320, 1024, 1024, 256, 256, 1024, 1024, 256, 256, 1024, 1024, 256, 256, 1024, 1024, 256, 256, 1024, 1024, 192, 192, 1024] |

Table 5: The original width and the adjusted width with our width-wise streamlining technique for each layer in VGG and ResNet Models.

## B.2 Training Details

For all the models we trained in this paper, we train each model with SGD, where the learning rate is firstly set at $0.1$, momentum is set at $0.9$ and weight decay is set at $0.0001$. For models trained on CIFAR datasets, the model is trained for 160 epochs, and the learning rate decays at 80-th epoch and 120-th epoch by $0.1$. For models trained on TinyImageNet, the model is trained for 90 epochs and the learning rate is decayed at the 30-th and 60-th epoch by $0.1$. Note that the results reported in Table 3 are averaged 10 runs where we train each model for 120 epochs and the learning rate is decayed at the 40-th and 80-th epoch by $0.1$. For the proposed width streamlining method, we mainly conduct experiments with supervised learning, we hope our method could further expand used in more scenarios such as self-supervised learning [40, 39, 38].

