# OpenReview forum: "Unveiling The Matthew Effect Across Channels: Assessing Layer Width Sufficiency via Weight Norm Variance"
_NeurIPS.cc/2024/Conference — NeurIPS 2024 poster_

### Official Review · Reviewer_nGxF · 2024-07-12

**Soundness:** 4
**Presentation:** 3
**Contribution:** 4
**Rating:** 8
**Confidence:** 5

**Summary:**

The paper studies the effects of layer widths on neural network performance. By studying the affects of the weight norm in different channels, the work discovers several distinct stages of training, which are apparent across different modalities and architectures. The authors also show how the insights could help in improving network performance with the same computational costs.

**Strengths:**

The paper has great merit, and discovers interesting dynamics in neural network training. Further, the dynamics are consistent, and come from an intuitive intuition of NN training. It is exhibited across different architectures and datasets, with an ability to also improve networks using these insights.

**Weaknesses:**

The paper is really great! It could benefit from more experiments and archs, but the current experiments are convincing.

**Questions:**

Some possible interesting references which could be relevant:

1. https://proceedings.neurips.cc/paper_files/paper/2023/hash/b63ad8c24354b0e5bcb7aea16490beab-Abstract-Conference.html

---

> ### Author Rebuttal · Authors · 2024-08-06
>
> Dear reviewer nGxF,
>
> Thank you very much for your thoughtful and valuable feedback. Your recognition of the intuitiveness of our findings and the potential for our insights to further improve neural networks is very encouraging and motivating for us. We are committed to further advancing this line of research by conducting more experiments and arches.
>
> > Some possible interesting references which could be relevant.
>
> Thank you for introducing the references, which are indeed interesting in providing insights into the representations learned in self-supervised learning.
>
> Once again, thank you for your valuable comments and support. We are more than happy to respond to any further questions.

---

### Official Review · Reviewer_nV2M · 2024-07-14

**Soundness:** 3
**Presentation:** 4
**Contribution:** 3
**Rating:** 6
**Confidence:** 4

**Summary:**

The paper proposes a method to optimize neural network layer width by analyzing the variance of weight norms across channels during training. This approach helps determine if a layer is sufficiently wide. Empirical validation shows that adjusting layer widths based on these patterns can reduce parameters and improve performance across various models and datasets.

**Strengths:**

1. This paper proposes a novel way to assess layer width sufficiency using weight norm variance.

2. The paper shows robust experimental evidence across multiple datasets and model architectures.

3. Besides the application on the model width optimization, the paper also offers a deeper understanding of training dynamics through the identification of distinct patterns in weight norm variance, which could be valuable for other related areas.

**Weaknesses:**

1. The layer width optimization is related to channel pruning and NAS-based channel number search, which have achieved significant success in finding optimal layer widths. I think these methods should be discussed and compared in the paper.

2. Measuring the weight norm would introduce additional computation cost, can the authors discuss the complexity of the method and report how much additional time does the method introduce?

3. Can the authors provide more detailed and theoretical analysis on the choice of the metric? For example, there are other metrics such as gradient norm, Hessian matrix, and absolute weight value to evaluate the weight importance and training statistics, why the proposed metric is better?

**Questions:**

See Weaknesses.

**Limitations:**

The limitations have been adequately discussed in the paper.

---

> ### Author Rebuttal · Authors · 2024-08-06
>
> Dear reviewer nv2M,
>
> Thank you for your valuable feedback. We will address each of your concerns and questions in the following section.
>
> > W1: The layer width optimization is related to channel pruning and NAS-based channel number search, which have achieved significant success in finding optimal layer widths. I think these methods should be discussed and compared in the paper
>
> Thanks for the suggestion. We will add more discussion and comparisons to our paper.  Generally, our approach differs from the two kinds of methods mentioned above.
> * For channel pruning, the objective is to reduce the computational cost, which differs from the objective of this paper. Notably, most pruning methods would manually set a pruning ratio for each layer.
> * For NAS-based search, the model width is searched over a model structure space. However, in this paper, we want to provide a more principled indicator for whether one layer in a network is sufficiently wide.
>
> > W2: Measuring the weight norm would introduce additional computation costs. Can the authors discuss the complexity of the method and report how much additional time the method introduces?
>
> The computation cost for measuring the weight norm is less than that of inferencing once with the model. We report the time used for measuring the weight norm on the CPU (AMD EPYC 7302 16-Core Processor).
> | model    | time used(s)       |
> | -------- | ------------------ |
> | VGG16    | 0.4580$\pm$ 0.3560 |
> | ResNet18 | 0.3150$\pm$ 0.2433 |
>
> The results are averaged over 10 runs, and we report the mean and the std of the results. Therefore, the time required to measure the weight norm is neglectable. We will add a more comprehensive computational cost analysis in our paper.
>
> > W3: Can the authors provide a more detailed and theoretical analysis of the choice of the metric? For example, there are other metrics such as gradient norm, Hessian matrix, and absolute weight value to evaluate the weight importance and training statistics. Why is the proposed metric better?
>
> Since we are focusing on the channels in one layer, the output of each channel is combined (mostly added up) and becomes the layer's output. For one channel, the weight norm generally determines how influential the channels are to the layer's output, whereas a larger weight norm produces a larger output.
>
> For other metrics, we have shown in Sec. 3.1 that the gradient norm is positively correlated to the weight norm, which also requires backward propagation on certain losses to be measured. While absolute weight value and Hessian matrix do not fit our purpose.
>
> We hope our rebuttal could address your concerns. We are looking forward to your reply. Thanks for your time and effort during the reviewing process. We are more than happy to answer any further questions.

---

### Official Review · Reviewer_iwdV · 2024-07-16

**Soundness:** 2
**Presentation:** 3
**Contribution:** 2
**Rating:** 4
**Confidence:** 5

**Summary:**

This paper tries to address an issue in deep neural networks: the trade-off between computational cost and performance, particularly focusing on the width of each layer. Traditionally, layer widths in neural networks are determined empirically or through extensive searches. The authors propose a novel approach by examining the variance of weight norms across different channels to determine if a layer is sufficiently wide.

**Strengths:**

1. The authors identify patterns regarding the variance of weight norm between different channels during training. Some layers exhibit an "increase to saturate" (IS) pattern and other layers show a "decrease to saturate" (DS) pattern. These patterns have some connection regarding the inter-channel similarities given a layer.
2. They redesign the width of classical CNN models like ResNets and VGGs according to their findings. And the empirical results provide some support for their arguments.

**Weaknesses:**

1. The empirical justification of the proposed method is somehow not convincing enough. From Table 2, we can see that channels within layers with IS patterns are also hard to merge, like layers 8 to 11 in Table 2. From Figure 12 in the Appendix, we can see that the similarity gradually decreased first and then increased for middle layers with DS patterns, and it is hard to say there is a hard cutoff.
2. Suppose the proposed argument is correct, then it seems challenging to use this method in practice. The width of each layer depends on the IS or DS pattern of the original model, so you need to train the original model first and then the model with the streamline width. This always increases the total training cost especially when scaling up the model.
3. The authors claim that they have similar observations for ViTs in lines 58-62, but no results are given. Since the original ViT has a uniform width, I am wondering whether the arguments still hold for models with uniform width.
4. The experiment setup is not very convincing. For CIFAR-10/100 scales of datasets, ResNet-18 or ResNet-50 are large and they have many redundant parameters. In the original ResNet paper, they used different architectures for CIFAR datasets with a much smaller number of parameters (0.27M to 1.7M with ResNet-20 to ResNet-110), please see Table 6 in "Deep Residual Learning for Image Recognition". The performance of ResNets and VGGs is much lower than public baselines for CIFAR-10 and CIFAR-100. Please see these two repo: https://github.com/weiaicunzai/pytorch-cifar100 and https://github.com/kuangliu/pytorch-cifar.

**Questions:**

Please see the weakness.

**Limitations:**

Addressed in conclusion.

---

> ### Author Rebuttal · Authors · 2024-08-06
>
> Dear reviewer iwdV,
>
> Thank you for your valuable feedback. We address each of your concerns in the following.
>
> > W1: The empirical justification of the proposed method is somehow not convincing enough. It is hard to say there is a hard cutoff.
>
> With all due respect, our results indicate that there is no hard cutoff between the DS pattern and the IS pattern. As shown in Sec. 3.2, as the layer width increases, the pattern gradually changes from the DS pattern to the IS pattern. Similar to Fig. 9 and Fig. 12 in the Appendix, the pattern and the similarity gradually change from IS to DS and from DS to IS again. In our hypothesis, the perfect width would be that the pattern is at the edge between the DS and IS pattern.
>
> > W2: Suppose the proposed argument is correct, then it seems challenging to use this method in practice.
>
> Generally, whether the weight norm variance follows the IS or DS pattern is shown way before convergence, which makes a more efficient method possible.
> Still, we agree that finding a more efficient method is challenging. However, the community has witnessed the development of more efficient methods for finding lottery tickets [1], where retraining is also required in the first paper. We hope our findings in this paper could innovate future works in adjusting layer width more efficiently.
>
> > W3: The authors claim that they have similar observations for ViTs in lines 58-62, but no results are given.
>
> Sorry for the confusion. The results regarding ViTs are presented in Sec. 3.2 and Fig. 3 [lines 153- 167]. As shown in Fig. 3, as we change the width of the MLP in a small ViT, the narrower MLP show a DS pattern, while the wider MLP shows an IS pattern. This phenomenon aligns with our observations on GCN, GRU, and CNNs, in that wider layers show the IS pattern while narrow layers show the DS pattern.
>
> > W4: In the original ResNet paper, they used different architectures for CIFAR datasets with a much smaller number of parameters (0.27M to 1.7M with ResNet-20 to ResNet-110).
>
> We conduct experiments with ResNet-20 on CIFAR10.  Since it is a much smaller network, we show that the layers in the first two blocks follow the DS pattern, and only the layers in the third block follow the IS pattern. By widening the small ResNet-20 by 8 times, we show that the layers in the first block and the third block show the IS pattern, and the layers in the second block show the DS pattern.
>
> Since these results are in figures, they are in the PDF attached to the joint rebuttal.
>
> > W5: The performance of ResNets and VGGs is much lower than public baselines for CIFAR-10 and CIFAR-100.
>
> The main difference between our implementation and the two mentioned repos is that we only use random flip in the training data preprocess while the two repos use normalization, and the repo for CIFAR100 uses random rotation in the data preprocessing.
>
> We report the results using the code of the two mentioned repos:
>
> **CIFAR10**
> | Model               | accuracy       |
> | ------------------- | -------------- |
> | VGG16 | 94.03$\pm$ 0.09|
> | VGG16 streamline    | __94.19$\pm$ 0.10__ |
> |ResNet18| 95.12$\pm$ 0.12|
> | ResNet18 streamline | __95.60$\pm$ 0.09__ |
> **CIFAR100**
>
> | Model | accuracy  |
> | ------------------- | --------------- |
> | VGG16 | 71.25$\pm$ 0.25|
> | VGG16 streamline    | __71.90$\pm$ 0.23__ |
> |ResNet18 | 74.33 $\pm$ 0.21|
> | ResNet18 streamline | __74.94$\pm$ 0.23__ |
>
> Each result is averaged over $5$ runs. We report the mean and std of the accuracy.
>
> We hope our rebuttal addresses your concerns, and we look forward to your reply. Thanks for your time and effort during the reviewing process. We are more than happy to answer any further questions.
>
> [1] Frankle, Jonathan, and Michael Carbin. "The lottery ticket hypothesis: Finding sparse, trainable neural networks." ICLR 2019

---

> > ### Author Response · Authors · 2024-08-12
> > **Look forward to further discussion**
> >
> > Dear reviewer iwdV
> >
> > We hope this message finds you well. We appreciate your valuable feedback and have tried our best to address each point of your concerns in our rebuttal. Since the discussion period is approaching its end, we are eager to hear from you, whether our rebuttal has addressed your concerns. Please feel free to comment on our rebuttal if you have further questions or comments. We are more than happy to respond to any further comments. Thank you for your commitment to the review process.
> >
> > Best regards,
> >
> > Authors

---

> > ### Comment · Reviewer_iwdV · 2024-08-12
> >
> > I appreciate the efforts of the authors in the rebuttal. The rebuttal addressed some of the concerns. However, I believe there is more space to make the experiment more comprehensive on larger-scale datasets, models, and different kinds of architecture beyond CNNs (as stated in W2). As a result, I will keep my score.

---

> > > ### Author Response · Authors · 2024-08-12
> > > **Thank you for the comment and here are our response**
> > >
> > > Dear reviewer iwdV,
> > >
> > > Thank you for your feedback and we regret to hear that our rebuttal is not to your satisfaction. The following is our response to your further comment.
> > >
> > > As you mentioned "different kinds of architecture beyond CNNs", with all due respect, we want to emphasize that proposing a streamlining technique is only a small part of this paper.  As the title of this paper indicates, we mainly identify the weight norm variance pattern regarding the layer width, **where we provide theoretical analysis in Sec. 3.1 and extensive experimental results on RNN, GNN, Transformer, and CNN in Sec. 3.2.**
> > >
> > > In utilizing the identified pattern, besides the proposed method, we investigate the training dynamic of widely used CNNs with extensive experimental results providing insights such as there are three stages during the training procedure. We have also provided our analysis regarding each stage of training.
> > >
> > > The proposed method and corresponding experiments in Sec.5 are our attempts to provide a simple sample using the identified pattern to guide the network design and further validate the observations and insights provided in this paper. **In believing the novelty of the insights and the new perspective provided in this paper, we agree with the comment that "there is more space" for more effective methods. We sincerely hope our paper can innovate future works and contribute to the community in the attempt to follow the footsteps of many excellent previous works (Best papers) [1, 2] in providing insights and innovating further applications.**
> > >
> > > Thank you once again for your valuable time and feedback. We hope the identified pattern and provided insights in this paper could also be taken into consideration, as well as the proposed method.
> > >
> > > Best regards,
> > >
> > > Authors
> > >
> > > [1] Frankle, Jonathan, and Michael Carbin. "The lottery ticket hypothesis: Finding sparse, trainable neural networks." ICLR 2019, Best Paper.
> > >
> > > [2] Schaeffer, Rylan, Brando Miranda, and Sanmi Koyejo. "Are emergent abilities of large language models a mirage?." NeurIPS 2023, Best Paper.

---

### Official Review · Reviewer_Q6z7 · 2024-07-21

**Soundness:** 3
**Presentation:** 3
**Contribution:** 3
**Rating:** 6
**Confidence:** 4

**Summary:**

This paper investigates the relationship between the differences in weight norms across channels and the adequacy of layer widths. The authors suggest that knowing these patterns(IS/DS) can help set layer widths better, leading to better resource use, fewer parameters, and improved performance in different network designs. The experiment shows that narrow-wide-narrow streamline network would boost the performance.

**Strengths:**

Originality: Yes, this paper is the first work that studies the width of network from the perspective of variance of weight norm.

Quality. Good, clear figures and enough experiments.

Clarity. Good, the organization of the paper is well

Significance. Yes, it provide a new metric to adjust the width of the network.

**Weaknesses:**

To determine the appropriate width, your method should involve training the model and observing various patterns of weight norm variance to see if the width is sufficient. However, retraining the model to decide on the width can be time-consuming. Considering that IS/DS indicate whether layers learn similar neurons, could we use the cosine similarity of a pretrained model to directly assess if the width is sufficient?

**Questions:**

1. The experiment includes both ResNet and VGG16. How should we define the weight norm of convolutional layers? Chapter 3 provides a simple analysis of the Matthew effect between similar channels. Is there any theoretical analysis for convolutional layers?
2. DS patterns suggest less similar neurons, indicating we should increase the width. IS patterns suggest redundant neurons, meaning we can decrease the width. In practice, why don't we adjust the width exactly according to the IS/DS patterns? For example, in the appendix, Figure 9 shows IS/DS patterns for 13 layers of the VGG network. Layers 2, 3, 4, 5, 6, and 7 show DS patterns, so we should increase the width of these layers. However, in Table 13, only the widths of layers 2, 4, and 7 are increased. Would adjusting all the layers accordingly result in better performance?
3. How do we determine the exact width of a layer based on the DS pattern? Is there an empirical boundary for the width of layers? I noticed that in the third stage of weight norm variance, DS patterns exhibit different drop ratios. For instance, in Figure 9, layers 2, 3, and 7 drop to a higher level, while layers 4, 5, and 6 drop to a lower level. Does this indicate that we should increase the width more for layers 4, 5, and 6?
4. In table3, after adjusting to streamline width, all networks would have a higher FLOPs than origin. Would the performance improvement due the FLOPs increase?

**Limitations:**

No apparent limitations.

---

> ### Author Rebuttal · Authors · 2024-08-06
>
> Dear reviewer Q6z7,
>
> Thank you for your valuable feedback. We address each of your concerns and questions in the following.
>
>
> > W1: Retraining is time-consuming. Considering that IS/DS indicates whether layers learn similar neurons, could we use the cosine similarity of a pre-trained model to directly assess if the width is sufficient?
>
> Yes, for pre-trained models, using cosine similarity could be a more efficient way to assess if the width is sufficient. However, when it comes to inspecting a model on a new dataset, using weight norm variance indicates whether the layer width is sufficient way before convergence.
>
> > Q1: The experiment includes both ResNet and VGG16. How should we define the weight norm of convolutional layers? Chapter 3 provides a simple analysis of the Matthew effect between similar channels. Is there any theoretical analysis for convolutional layers?
>
> The difference between a convolutional layer and a linear layer is that the scalar multiplication in a linear layer becomes a convolution with the kernel as a matrix. Like in a linear layer, the larger kernel produces a larger output. In practice, for the weight norm, we reshape the kernel matrix into a vector and use the l2 norm of the vector to further calculate the weight norm of each channel. Thanks for the good question, we will add more analysis in our paper and clarify it.
>
> > Q2: DS patterns suggest less similar neurons, indicating we should increase the width. IS patterns suggest redundant neurons, meaning we can decrease the width. In practice, why don't we adjust the width exactly according to the IS/DS patterns?
>
> Yes, we can adjust the width exactly according to the pattern. In the paper, we manually adjust the layer width, and we want to make the model into a streamlined shape while the layer width is neat. To answer the question, we have made a VGG16_muted version 2 and tested it. The result is in the response to Q4.
>
> > Q3: How do we determine the exact width of a layer based on the DS pattern? Is there an empirical boundary for the width of layers? I noticed that in the third stage of weight norm variance, DS patterns exhibit different drop ratios. For instance, in Figure 9, layers 2, 3, and 7 drop to a higher level, while layers 4, 5, and 6 drop to a lower level. Does this indicate that we should increase the width more for layers 4, 5, and 6?
>
> You are right. According to results in Sec. 3.2, the DS pattern exhibits different drop ratios. As we increase the width, the pattern moves from DS to IS. The perfect scenario would be finding the smallest width leading to an IS pattern.
>
> > Q4: In Table 3, after adjusting to streamline width, all networks would have higher FLOPs than the origin. Would the performance improvement due to the FLOPs increase?
>
> Since the shallow layers take much more calculation than the deep layers in CNN, for the streamlined networks, the increase in FLOPs is not significant compared to the decrease in the number of parameters.
> We have also made a new version of VGG16_muted with lower FLOPs than the original VGG16 (99.68%).
>
> | model           | FLOPs (%) | params (%) | CIFAR10 accuracy | CIFAR100 accuracy |
> | --------------- | --------- | ---------- | ---------------- | ----------------- |
> | VGG16           | 100%      | 100%       | 94.03$\pm$ 0.09  | 71.25$\pm$ 0.25   |
> | VGG16 steamline | 103.89%   | 61.55%     | __94.19$\pm$ 0.10__  | __71.90$\pm$ 0.23__   |
> | VGG16 streamline v2 |99.68%|   58.53%    |     94.16$\pm$ 0.11|71.68$\pm$ 0.29|
>
> For this experiment, we add normalization and random rotation in the preprocess of training data as suggested by reviewer iwdV. We will provide more experimental results and add them to our paper. Generally, VGG16 streamlined version 2 reduces both the FLOPs and the number of parameters while gaining performance increases.
>
> We hope our rebuttal addresses your concerns, and we look forward to your message. Please feel free to make any comments; we are more than happy to respond to any further questions.

---

### Author Rebuttal · Authors · 2024-08-06

Dear AC and Reviewers,

We sincerely thank you for the time and effort you dedicated to the reviewing process. We are delighted to hear that reviewers find the paper to be well-written (Q6z7, nV2M, nGxF), novel (Q6z7, nV2M), and interesting (nGxF). To further address the comments and questions posed by reviewers, we have also conducted additional experiments as required by the reviewers, including:
* Measuring the time required to calculate the weight norm.
* Train the networks with normalization added to the preprocessing of training data.
* Conduct experiments with smaller ResNet-20. (**The results in figures are provided in the attached PDF**)

For each reviewer, we have posed a rebuttal addressing the concerns. We look forward to your reply and are more than happy to respond to any further comments. Once again, thank you for your valuable comments and support.

---

### Comment · Area_Chair_utGb · 2024-08-09

Dear Reviewers,

I hope this message finds you well. I would like to take this opportunity to express my gratitude for the valuable feedback and insights you have provided during the review process. Your expertise and dedication are essential to maintaining the high standards of our community.

The authors have devoted considerable time and effort to addressing the concerns and suggestions you raised in your reviews. They have carefully crafted a detailed rebuttal, aiming to clarify any misunderstandings and incorporate your feedback into their manuscript.

I kindly ask that you take a thorough and thoughtful look at the authors' responses. Your assessment of their rebuttal is crucial in determining whether they have satisfactorily addressed the issues and concerns highlighted in your initial review.

Thank you once again for your hard work and commitment to advancing the quality of our scholarly community. Your contributions are greatly appreciated.

Best regards,
AC

---

### Author Response · Authors · 2024-08-11
**Look forward to further discussion**

Dear AC and Reviewers,

We sincerely appreciate your valuable feedback and the efforts you devoted to the reviewing process. Since the discussion is approaching its end, we look forward to your opinion on whether our rebuttal has addressed the concerns and issues in the initial review.

In the initial review, we are glad to find that the reviews are generally positive. We cherish your valuable feedback and the opportunity for us to have further discussions. We will keep on polishing our paper and are more than happy to respond to further comments.

Best regard,

Authors

---

### Decision · Program_Chairs · 2024-09-25

**Decision:**

Accept (poster)

**Comment:**

The paper proposes a method to optimize neural network layer width by analyzing the variance of weight norms across channels during training. This approach helps determine if a layer is sufficiently wide. Empirical validation shows that adjusting layer widths based on these patterns can reduce parameters and improve performance across various models and datasets.

Most reviewers find the paper is well-written, and the results are interesting.
Please make sure to incorporate the feedback in the camera-ready version, especially addressing the reviewers' concerns on the experimental setup and more comprehensive studies on larger datasets.